# Mizoribine Promotes Molecular Chaperone HSP60/HSP10 Complex Formation

**DOI:** 10.3390/ijms25126452

**Published:** 2024-06-12

**Authors:** Atsuko Miura, Yukihiko Narita, Taku Sugawara, Hiroaki Shimizu, Hideaki Itoh

**Affiliations:** 1Department of Neurosurgery, Akita University Graduate School of Medicine, Akita 010-8543, Japan; 2Department of Life Science, Akita Cerebrospinal and Cardiovascular Center, Akita 010-0874, Japan; 3Department of Applied Biological Chemistry, Graduate School of Agricultural and Life Sciences, The University of Tokyo, Tokyo 113-8657, Japan

**Keywords:** immunosuppressant, Mizoribine, molecular chaperone, HSP60–HSP10, IL-6

## Abstract

It has been reported that Mizoribine is an immunosuppressant used to suppress rejection in renal transplantation, nephrotic syndrome, lupus nephritis, and rheumatoid arthritis. The molecular chaperone HSP60 alone induces inflammatory cytokine IL-6 and the co-chaperone HSP10 alone inhibits IL-6 induction. HSP60 and HSP10 form a complex in the presence of ATP. We analyzed the effects of Mizoribine, which is structurally similar to ATP, on the structure and physiological functions of HSP60–HSP10 using Native/PAGE and transmission electron microscopy. At low concentrations of Mizoribine, no complex formation of HSP60–HSP10 was observed, nor was the expression of IL-6 affected. On the other hand, high concentrations of Mizoribine promoted HSP60–HSP10 complex formation and consequently suppressed IL-6 expression. Here, we propose a novel mechanism of immunosuppressive action of Mizoribine.

## 1. Introduction

Mizoribine is a class of disease-modifying antirheumatic drugs (DMARDs). Mizoribine is an antimetabolite that inhibits the purine synthesis system of nucleic acids and was developed in Japan in 1971 [1]. The chemical structural formulae of Mizoribine, ATP, and GTP are shown in Figure 1.

Mizoribine is an imidazole nucleotide that selectively inhibits inosine-5-monophosphate dehydrogenase, an enzyme in the de novo purine nucleoside synthesis system [2,3], resulting in the inhibition of T and B lymphocyte proliferation. Furthermore, recent studies suggest that Mizoribine directly prevents podocyte injury and maintains nephritic structure, decreasing urinary protein [4]. This immunosuppressive effect is thought to be caused by Mizoribine monophosphate, produced by an adenosine kinase reaction after intracellular uptake. Mizoribine monophosphate inhibits inosine monophosphate (IMP) dehydrogenase and guanosine monophosphate (GMP) synthase, resulting in depletion of intracellular GTP levels and suppressing T cell proliferation [5].

Mizoribine is used in the treatment of nephrotic syndrome, lupus nephritis, and rheumatoid arthritis [5,6]. Mizoribine is phosphorylated intracellularly and selectively and competitively inhibits inosine phosphate dehydrogenase (IMP dehydrogenase), the rate-limiting enzyme in the de novo pathway of lymphocyte nucleic acid synthesis. As a result, it exerts its immunosuppressive effect by arresting DNA synthesis in the S phase of cell division, thereby inhibiting the proliferation of T and B cells [7,8].

Human HSP60 promotes the expression of the adhesion molecules E-selectin, intercellular adhesion molecule (ICAM)-1, and vascular cell adhesion molecule (VCAM)-1 in vascular endothelial cells and also induces the secretion of interleukin-6 from vascular endothelial cells, smooth muscle cells, and macrophages [9,10]. In addition, bacterial HSP60 (also called GroEL) causes the expression of ICAM-1 and VCAM-1 in vascular endothelial cells. Chlamydial HSP60 promotes the expression of E-selectin, ICAM-1, and VCAM-1 in human vascular endothelial cells and activates the secretion of interleukin-6 from these cells, smooth muscle cells, and macrophages [11].

It has been reported that HSP60 alone induced the proinflammatory cytokine IL-6 in a dose-dependent manner [12]. On the other hand, the co-chaperone HSP10 of HSP60 alone was reported to inhibit the induction of IL-6 [13]. We have prepared a Mizoribine affinity column using an Epoxy-activated Sepharose 6B, analyzed the specific binding proteins to the molecular chaperone HSP60, and reported that Mizoribine inhibits its chaperone activity [14]. Based on the above findings, we investigated HSP60–HSP10 complex formation and IL-6 induction in the presence of Mizoribine and investigated the immunosuppressive mechanism of Mizoribine.

## 2. Results

The chemical structural formulae of Mizoribine, ATP, and GTP are shown in Figure 1. Since Mizoribine binds specifically to the molecular chaperone HSP60, we analyzed the effect of Mizoribine on the complex formation between HSP60 and HSP10. Purified human HSP60 and HSP10 were analyzed using SDS/PAGE (Figure 2A). Their molecular mass was analyzed on Native/PAGE (Figure 2B). In the absence of ATP, HSP60 and HSP10 each form a heptameric structure (Figure 2B). In contrast, football-type HSP60_14_/HSP10_14_ or bullet-type HSP60_14_-HSP10_7_ have been reported in the presence of ATP [15,16,17,18]. We analyzed the molecular mass of the HSP60 and HSP10 complex on Native/PAGE in a Mizoribine dose-dependent manner (Figure 3A). At more than 10 mM Mizoribine, a protein band was detected at a position larger than the 720 kDa molecular marker. These results suggest the formation of HSP60_14_/HSP10_14_ (double-ring complex of HSP60 and HSP10 each). In the presence of Mizoribine below 10 mM, HSP60 and HSP10 were suggested to be in a heptameric structure. We observed HSP60 and HSP10 complex structures with Mizoribine by other methods. We used TEM to analyze the structure of the HSP60–HSP10 complex. When in the absence of Mizoribine, HSP60 was essentially a single-ring structure of a heptamer (Figure 3B). On the other hand, in the presence of 10 mM of Mizoribine, HSP60–HSP10 was mostly observed as a football-type complex of HSP60_14_–HSP10_14_, rarely as a bullet-type complex of HSP60_7_–HSP10_7_ or as a dimer of only HSP60_7_ (Figure 3C). In the average of 100 molecular observations in a given area, HSP60 was a single-ring structure in the absence of Mizoribine. On the other hand, the structures of HSP60 and HSP10 in the presence of Mizoribine were HSP60_14_–HSP10_14_ (football type), HSP60_7_–HSP10_7_ (bullet type), and HSP60_7_ (single ring structure) with the ratio of 66:30:4, respectively (Figure 3D).

We quantified the mRNA levels of the proinflammatory cytokine IL-6. Mizoribine was added to the Caco2/THP-1 transwell system. IL-6 mRNA was quantified at 3 h after Mizoribine administration (Figure 4A,B). The mRNA expression level of IL-6 was significantly decreased at 10 and 20 mM of Mizoribine. We then fixed the concentration of Mizoribine at 10 mM and analyzed the protein level of IL-6 over time at 0, 3, and 6 h (Figure 4C). In the immunoblot in Figure 4C, β-actin and IL-6 antibodies reacted simultaneously. Although several protein bands reacting with these antibodies were detected, based on the manufacturer’s datasheet, bands of 45 kDa for β-actin and approximately 20 kDa for IL-6 were detected. The amount of IL-6 protein at 3 and 6 h after administration of Mizoribine was drastically reduced to about 10% of that in controls (Figure 4D). Thus, Mizoribine suppresses the expression of IL-6, an inflammatory cytokine.

## 3. Discussion

Mizorivine, isolated from *Eupenicillium brefeldianum*, is an antimetabolite that inhibits the purine synthesis system of nucleic acids, a drug developed in Japan in 1971. Mizoribine was initially developed to prevent rejection in renal transplantation; it was later added as a treatment for nephrotic syndrome caused by primary glomerulonephritis, lupus nephritis, and rheumatoid arthritis. Mizoribin is used to prevent rejection after kidney and other organ transplants and is also effective for diseases involving the immune system, such as collagen disease, nephrosis, and rheumatoid arthritis [19,20,21,22].

MZR is an imidazole nucleotide that inhibits purine synthesis and T and B lymphocyte proliferation [2,3]. These actions are less myelosuppressive and less hepatotoxic [23,24].

Molecular chaperone HSP60 has been shown to trigger an innate immune response that initiates the earliest yet reversible inflammatory phase, as well as an adaptive immune response [25,26,27,28]. Normal adult mice have a large subset of pre-existing peripheral lymphoid gamma-delta T cells that are responsive to molecular chaperone HSP60. All of its members have been reported to interact with one domain of this common HSP60 [29]. HSP60 has been reported to stimulate both adipocytes and their precursors to release IL-6, MCP-1, and KC in a dose-dependent manner and HSP60 was recently identified as an immune-mediating molecule that plays a central role in obesity-related inflammatory processes [10]. HSP10, originally identified as an early pregnancy factor present in early pregnancy serum, has shown immunosuppressive activity in experimental autoimmune encephalomyelitis, delayed hypersensitivity, and graft rejection models [30,31]. Studies using endotoxin-free recombinant HSP10 have reported that it reduces the secretion of TNF-α and IL-6, whose expression is induced by LPS. It has been suggested that HSP10 likely suppresses inflammation-induced responses by associating with extracellular HSP60 [8,11,12,13,14]. We have previously demonstrated that the specific binding protein for Mizoribine is HSP60 using a Mizoribine affinity column [14].

Interleukin-6 (IL-6) is a multifaceted inflammatory cytokine that is transiently produced by tissue injury or infection. Although its expression is regulated by tightly controlled transcriptional and post-transcriptional mechanisms, dysregulation of the continuous production of IL-6 in inflammatory conditions can adversely affect immune cells. Molecular evidence demonstrates a detrimental turn of IL-6 trans-signaling in the pathogenesis of one such autoimmune joint disease, rheumatoid arthritis (RA). The significant increase in IL-6 in rheumatoid arthritis, along with multiple growth factors released primarily by synovial-like fibroblasts (FLS) and macrophages, is essential for clinical disease progression. Due to its pathogenic nature, the blockade of IL-6, which mediates inflammation and situation-driven signaling cassettes, may be a powerful target in therapeutic interventions for RA [32].

IL-6 inhibitors are among the major biological agents used for rheumatoid arthritis. The signaling mechanism of IL-6 is that IL-6 binds to the membrane-bound IL-6 receptor (IL-6R) and the soluble IL-6 receptor (SIL-6R), and then signals are transduced to the nucleus to initiate transcription of the genetic code. IL-6 inhibitors are drugs that exert their antirheumatic effects by blocking IL-6 signaling. Currently, two drugs are available, Tocilizumab (also known as Atlizumab) and Sarilumab (also known as Kevzara), which have antibody activity against the IL-6 receptor [33].

It has been reported that Mizoribine could suppress IL-6 release in vitro and may exert its efficacy on IgA nephropathy [34]. Mizoribine was able to inhibit the spontaneous production of IL-6 by fresh rheumatoid synovial cells (RSC) in a dose–response fashion [35]. Thus, Mizoribine has been reported to inhibit the production of the anti-inflammatory cytokine IL-6. We focused on the inflammatory cytokine IL-6 induction ability of Mizoribine concerning its immunosuppressive effect on the promotion of HSP60 and HSP10 complex formation. Above 10 mM, Mizoribine suppressed the induction of IL-6 both at the mRNA and protein levels. Furthermore, in the presence of less than 10 mM Mizoribine, HSP60 was a single-ring structure, whereas, in the presence of 10 mM Mizoribine, 96% of HSP60 was of the football and bullet types. The inhibition of IL-6 induction in the presence of more than 10 mM Mizoribine and the formation of HSP60–HSP10 complexes were found to be closely correlated.

Mizoribine is less effective and slower-acting than other DMARDs. It is based on dosing concentrations of Mizoribone between 0.5 and 3 μg/mL. It is speculated that higher doses may be required than the current clinical dose of 2–5 mg/kg/day [26], otherwise, 5–10 mg/kg per day twice a week for more than 24 months. However, the maximum daily dosage at this time is 500 mg/kg/day [36]. This concentration of Mizoribine in vivo is 1.93 mM. Thus, high doses of Mizoribine are recommended in clinical practice.

In the present in vitro study, a low concentration of Mizoribine did not affect IL-6 induction and the structure of HSP60 remained in the shingles. Still, high concentrations of Mizoribine promoted HSP60–HSP10 complex formation and consequently inhibited IL-6 expression induction, showing immunosuppressive effects.

With regard to the difference between clinical and cellular concentrations of Mizoribine, we speculate as follows: pharmacokinetics in vivo are always effective in the bloodstream. In other words, the blood concentration of the drug changes over the time of administration. On the other hand, in cell-based drug experiments, we speculate that a higher concentration than the in vivo concentration is necessary because there is no blood flow. Therefore, the concentrations of Mizoribine in this study were several times higher than the clinically used concentrations. The novel immunosuppressive mechanism of Mizoribine is shown in Figure 5.

## 4. Materials and Methods

### 4.1. Materials

Mizoribine was obtained from Asahi Kasei Pharma (Tokyo, Japan). Phorbol12-myristate13-acetate (PMA) and Cholecalciferol (Vitamin D_3_) were obtained from Tokyo Chemical Industry Co., Ltd., Tokyo, Japan. Professor Shinichi Yokota, Sapporo Medical University, Japan, kindly provided human colon cancer Caco2 cells. Human acute monocytic leukemia THP-1 cells were obtained from RIKEN Cell Bank, Japan. The medium (DMEM and RPMI 1640) and antibiotics (penicillin–streptomycin mixed solution) were purchased from Nacalai Tesque Inc., Kyoto, Japan. FBS was purchased from Serena Europe GmbH (Pessin, Germany). Millicell, and 6-Well Hanging Inserts 0.4 μm PET (Cell Culture Inserts) were obtained from Merck (Darmstadt, Germany). Multiwell (6 wells) was purchased from Falcon (Glendale, AZ, USA). Millicell-ERS was purchased from Merck. Beta-actin rabbit polyclonal antibody (20536-1-AP) and IL-6 Rabbit Polyclonal antibody (21865-1-AP) were obtained from Proteinteck (Rosemont, IL, USA). An anti-rabbit AP IgG was purchased from BioRad Laboratories (Hercules, CA, USA). BCIP-NBT solution, Nonidet P-40, and Phenylmethylsulfonyl fluoride (PMSF) were purchased from Nacalai Tesque. Super Script III kit was purchased from ThermoFisher Scientific (Waltham, MA, USA). PrimeSTAR GXL Polymerase was purchased from Takara-Bio (Shiga, Japan).

### 4.2. Structural Analysis of Human HSP60 and HSP10

SDS/PAGE and immunoblotting were conducted according to Laemmli [37] (Laemmli, 1970). Human HSP60 and HSP10 were purified as described previously [16,17,18]. The molecular mass of the HSP60 and HSP10 in the absence or presence of Mizoribine (1, 5, 10, 15, and 20 mM) was analyzed using a Blue NativePAGE™ 3–12% Bis-Tris Gel (Invitrogen; Carlsbad, CA, USA) as described previously [18]. After electrophoresis, gels were stained with 0.1% Coomassie Brilliant Blue R-250 in a mixture of 25% isopropyl alcohol and 10% acetic acid; they were destained with 10% isopropyl alcohol and 10% acetic acid. The structure of HSP60 and HSP10 in the absence or presence of 10 mM Mizoribine was performed using transmission electron microscopy as described previously [16,17,18]. The structure of HSP60 and HSP10 was obtained by observing 100 molecules in a given area and averaging *n* = 5.

### 4.3. Cell Culture and Cell Differentiation

Cell culture and cell differentiation were followed as described previously [38]. Caco2 cells derived from human colon cancer (kindly provided by Professor Dr. Shinichi Yokota, Sapporo Medical University, Sapporo, Japan) and THP-1 derived from human acute monocytic leukemia (obtained from RIKEN Cell Bank, Tsukuba, Japan) were used in this study. Caco2 cells were cultured in a 5% CO_2_ incubator in Dulbecco’s converted Eagle’s medium (DMEM medium, Nacalai Tesque Inc., Kyoto, Japan) containing 5% FBS (Biological industries, Kibbutz Beit-Haemek, Israel) and 0.2% penicillin–streptomycin. THP-1 cells were cultured in RPMI 1640 medium (Nacalai Tesque Inc., Kyoto, Japan) containing 10% FBS in a 5% CO_2_ incubator. Caco2 cells were seeded on Cell Culture Inserts (Millicell, 6-Well Hanging Inserts 0.4 μm PET, Merck, Darmstadt, Germany) and cultured and differentiated in a medium containing 5.0 mM butyric acid for 4 days. The cell differentiation was measured by transepithelial electrical resistance (TEER) using Millicell-ERS (Merk) and differentiated cells with a TEER value above 400 Ω × cm^2^. THP-1 cells were seeded in multiwell (6 well, Glendale, AZ, USA) and cultured and differentiated for 3 days in a medium containing 1000-fold diluted Phorbol12-myristate13-acetate (PMA) and 2600-fold diluted Cholecalciferol (Tokyo Chemical Iindustry Co., Ltd., Tokyo, Japan). After each differentiation, Caco2 and THP-1 cells were co-cultured in a transwell.

### 4.4. Analysis of mRNA Levels and Protein Levels of IL-6

For analysis of IL-6 mRNA, 1, 5, 10, and 20 mM of Mizoribine was added to the upper insert of the Transwell system and incubated at 37 °C in the presence of 5% CO_2_ for 3 h. PBS was added as a control. Total RNA was isolated from THP-1 cells using RNeasy Mini Kit (Qiagen, Hilden, Germany). According to the manufacturer’s instructions, RNA (1 µg) was used for synthesizing cDNA with a Super Script III kit (Invitrogen). RT-PCR was performed using PrimeSTAR GXL Polymerase (Takara-Bio, Shiga, Japan). The condition was as follows: initial denaturation at 94 °C for 2 min, followed by 40 amplification cycles (denaturation at 94 °C for 15 s, annealing at 55 °C and extension at 68 °C, each 30 s). The cDNAs were amplified by SimpliAmp (ThermoFisher Scientific, Waltham, MA, USA) with the following primers: IL-6 forward primer 5‘-ACACAGACAGCCACTCACC and reverse primer 5′-TACATTTGCCGAAGAGCC, β-Actin forward primer 5′-GCTCGTCGTCGACAACGGCTC and reverse primer 5′-CAAACATGATCTGGGTCATCTTCTC. PCR products were analyzed in 1% agarose gel and stained with ethidium bromide. The band ratios were quantified using Image J software ver. 1.53 and normalized by β-actin.

For protein analysis of IL-6, 10 mM Mizoribine was added to the upper inserts of the Transwell system and incubated at 37 °C in the presence of 5% CO_2_ for 3 and 6 h. PBS was added as a control. After incubation, THP-1 cells were then washed with PBS and lysed with buffer (10 mM Tris-HCl, pH 7.4, 0.15 M NaCl, 1% Nonidet P-40, 1 mM PMSF), then homogenized using Physcotron homogenizer NS-310E (Microtec Co., Ltd., Chiba, Japan). The homogenized cell lysate was centrifuged at 20,000× *g* for 10 min at 4 °C. After centrifugation, the supernatant was dissolved in 2× SDS sample buffer (FUJIFILM Wako Pure Chemical Corporation, Osaka, Japan). Samples were electrophoresed on 18% acrylamide slab gel with SDS. After electrophoresis, gels were stained with 0.1% Coomassie Brilliant Blue R-250 in a mixture of 25% isopropyl alcohol and 10% acetic acid; they were destained with 10% isopropyl alcohol and 10% acetic acid or by immunoblotting [39] using an antibody against IL-6 (Proteintech, Rosemont, IL, USA) or β-actin. The PVDF membrane was reacted with IL-6 and β-actin antibodies with an anti-rabbit AP IgG (BioRad). Samples were treated with BCIP-NBT solution (Nacalai Tesque). Total IL-6 or β-actin was calculated using ChemiDoc XRS (BioRad) and normalized by β-actin.

### 4.5. Statistical Analysis

Data were expressed as mean values + S.D. Statistical analysis was performed using the Student’s *t*-test. Differences were considered statistically significant with *p* < 0.05.

## Figures and Tables

**Figure 1 ijms-25-06452-f001:**
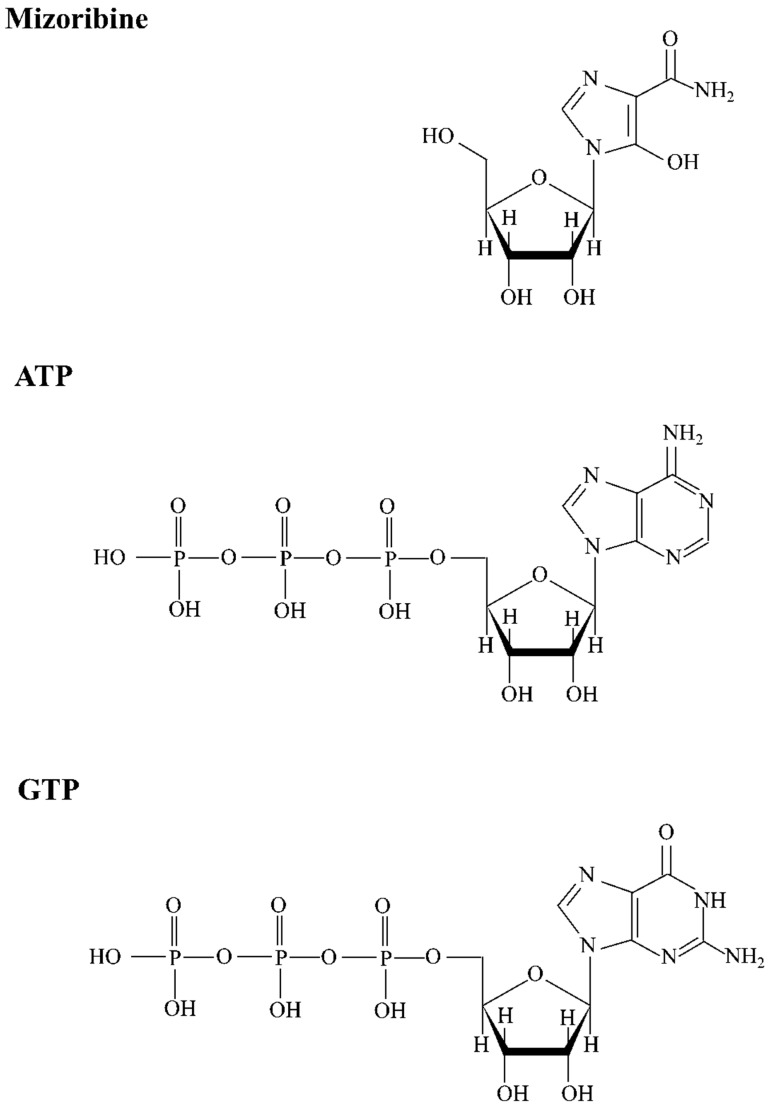
Chemical structure of Mizoribine, ATP, and GTP. The chemical structure of Mizoribine (4-carbamoyl-1-β D-ribofuranosylimidazo-lium-5-olatemonohydrate), ATP (adenosine triphosphate), and GTP (guanosine triphosphate) are shown.

**Figure 2 ijms-25-06452-f002:**
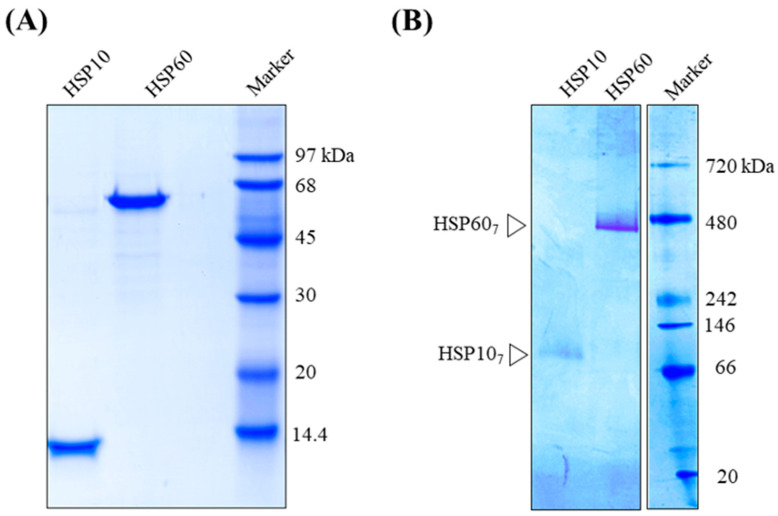
Human HSP60 and HSP10. Purified human HSP60 and HSP10 were analyzed by SDS-PAGE (12% gel) (**A**) and Native-PAGE (5–12% gradient gel) (**B**). HSP60_7_ and HSP10_7_ are indicated as open triangles, respectively.

**Figure 3 ijms-25-06452-f003:**
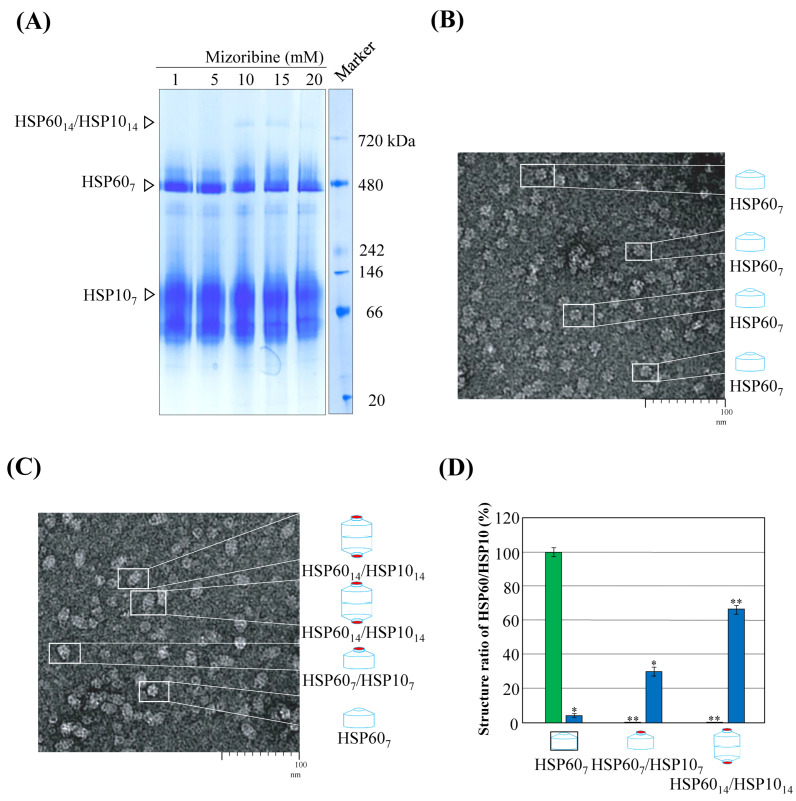
Effect of Mizoribine on the structure of HSP60 and HSP10. PBS or Mizoribine was added to the same molar ratio of the HSP60–HSP10 solution to a final concentration of 1, 5, 10, or 20 mM. The complex formation of HSP60 and HSP10 was analyzed by Native/PAGE (5–12% gradient gel) (**A**). HSP60_14_–HSP10_14_, HSP60_7_, and HSP10_7_ are indicated as open triangles, respectively. HSP60 and HSP10 (0.1 mM) were incubated for 10 min in the absence or presence of 10 mM Mizoribine. After incubation, these samples were treated with 2% uranyl acetate. Transmission electron microscopy (TEM) imaging of HSP60 in the absence (**B**) or presence (**C**) of Mizoribine. In the TEM images (**B**,**C**), the white rectangle is enlarged and the structural schematic of HSP60–HSP10 is shown on the right. (**D**) Classification of HSP60–HSP10 complex observed in the absence or presence of 10 mM Mizoribine based on (**B**,C). More than 500 side views of HSP60 complexes were counted for each experiment. The mean and standard deviations of three independent experiments are shown. The green and blue bars indicate the absence or presence of 10 mM Mizoribine, respectively. Data show means ± S.D. of three determinations. * *p* < 0.01; ** *p* < 0.001 compared to corresponding HSP60 complex in the absence of Mizoribine.

**Figure 4 ijms-25-06452-f004:**
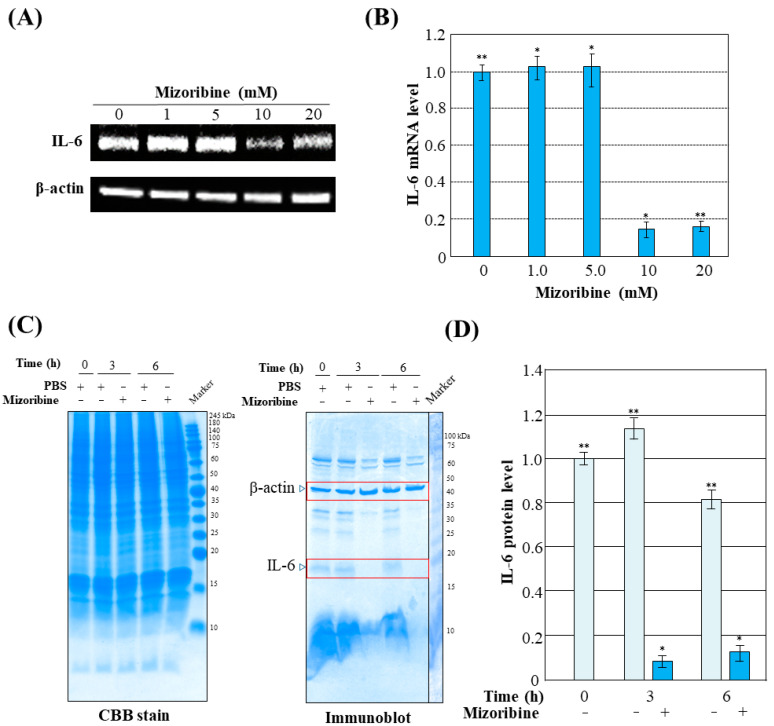
Effects of Mizoribine on induction of IL-6. Caco2 and THP-1 cells were co-cultured in Transwell. PBS or Mizoribine was added to cells for 3 h to a final concentration of 1, 5, 10, or 20 mM. IL-6 and β-actin were analyzed by RT-PCR (**A**). The IL-6/β-actin ratio based on (**A**) was quantified using Image J software ver. 1.53. Data show means ± S.D. of three determinations. * *p* < 0.01; ** *p* < 0.001 compared to corresponding *β*-actin control (**B**). PBS or Mizoribine was added to cells for 0, 3, and 6 h to a final concentration of 10 mM. The THP-1 cell lysate was analyzed by SDS-PAGE following immunoblotting using antibodies against IL-6 and β-actin (**C**). The IL-6/β-actin ratio based on (**C**) was quantified using ChemiDoc XRS. The signal in each lane of the immunoblot in (**C**) was quantified and graphed. The data are expressed as mean ± SD (*n* = 3) (**D**). Data show means ± S.D. of three determinations. * *p* < 0.01; ** *p* < 0.001 compared to corresponding β-actin control.

**Figure 5 ijms-25-06452-f005:**
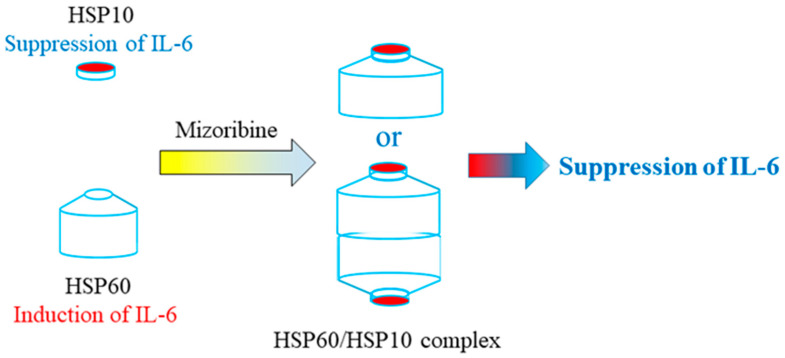
The novel immunosuppressive mechanism of Mizoribine. HSP10 alone inhibits the induction of IL-6 expression; HSP60 alone induces IL-6 expression. Mizoribine promotes complex formation between mHSP60 and HSP10; HSP60 and HSP10 complex inhibits induction of IL-6 expression. Mizoribine exerts its immunosuppressive effect by promoting the formation of molecular chaperone complexes.

## Data Availability

Data sharing does not apply to this paper, since all data are included within the main article.

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
