# Peer review of "Mizoribine Promotes Molecular Chaperone HSP60/HSP10 Complex Formation"

_ijms, 2024, doi:10.3390/ijms25126452_

Round 1

Reviewer 1 Report

Comments and Suggestions for Authors

Regulating proinflammatory cytokines in transplants or autoimmunity is critical for controlling unwanted immune reactions. People have used mizoribine, an immunosuppressive drug, to treat several ailments. Miura et al., have submitted the study entitled “Mizoribine Promotes Molecular Chaperone HSP60/HSP10 Complex Formation.”. The authors established that mizoribine influences the formation of the HSP60-HSP10 complex, thereby inhibiting the production of IL-6. The authors used a Caco2 and THP-1 cell-based assay to confirm the mizoribine-induced complex-mediated IL-6 inhibition.

I have the following comments:

The TEM results are not clear. CryoEM or other available technologies should present high-resolution images to double-confirm the results.

It is not clear where the interaction of HSP60 and HSP10 is located—extracellular, cytosolic, or nucleus. In inflammatory conditions, the nucleus hosts the majority of heat shock proteins. Wet lab experiments should address this.

The immunofluorescence assay using the colocalization technique should confirm the complex formation in the cells.

If you're submitting the manuscript for a full-length paper, you must answer the aforementioned queries. Otherwise, the manuscript will be ready for short communication.

Author Response

Comments and Suggestions for Authors

I have the following comments:

The TEM results are not clear. CryoEM or other available technologies should present high-resolution images to double-confirm the results.

It is not clear where the interaction of HSP60 and HSP10 is located—extracellular, cytosolic, or nucleus. In inflammatory conditions, the nucleus hosts the majority of heat shock proteins. Wet lab experiments should address this.

The immunofluorescence assay using the colocalization technique should confirm the complex formation in the cells.

If you're submitting the manuscript for a full-length paper, you must answer the aforementioned queries. Otherwise, the manuscript will be ready for short communication.

Thank you very much for reviewing our manuscript and for your valuable comments. We have sincerely responded to your points.

You are certainly right. However, since I do not have enough time to submit our Revise this time, we would like to choose short communication. I will leave the decision to the editor.

Text corrections are in red.

Text corrections are in red.

Reviewer 2 Report

Comments and Suggestions for Authors

In the submitted manuscript, the authors investigated and proposed a novel mechanism of the immunosuppressive action of Mizoribine. Upon review, the methods section is well-detailed and clearly described, ensuring reproducibility. The applied methods are appropriate for the study and robust in design. The data presented convincingly support the proposed results.

However, the introduction part need some improvement. It is too short. The structure of mizoribine should be moved to this section.

The majority of the figures should be reedited. One page figure is not acceptable.

The original source of the figures should be added to the data sharing section, or all the data should be published in Supplementary Materials.

Author Response

Comments and Suggestions for Authors

Thank you very much for reviewing our manuscript and for your valuable comments. We have sincerely responded to your points.

In the submitted manuscript, the authors investigated and proposed a novel mechanism of the immunosuppressive action of Mizoribine. Upon review, the methods section is well-detailed and clearly described, ensuring reproducibility. The applied methods are appropriate for the study and robust in design. The data presented convincingly support the proposed results.

However, the introduction part need some improvement. It is too short. The structure of mizoribine should be moved to this section.

We added an Introductory section and moved the structure of Mizoribine to this section, as suggested.

The majority of the figures should be reedited. One page figure is not acceptable.

The original source of the figures should be added to the data sharing section, or all the data should be published in Supplementary Materials.

The figure shows the reedited, as suggested.

Text corrections are in red.

Reviewer 3 Report

Comments and Suggestions for Authors

This work deals with the relevance of mizoribine in promoting the complex formation of the HSP60 with HSP10. Also, this compound can inhibit the production of IL6 in the cell culture system composed of Caco2 and THP1 cells.

The findings are of interest. Mizoribine is an immunosuppressive agent described several years ago. The effect on IL6 is already known for a long time. The authors stated that a novel mechanism of immune suppression mediated by mizoribine has been identified. Indeed, high concentrations of mizoribine promoted HSP60-HSP10 complex formation and consequently suppressed IL-6 expression.

This is the most important finding of this work. In other words, the mizoribine appears to favour a complex among the two different HSPs that reduces the production of IL6.

In the abstract, the authors stated that ATP induces the complex HSP60-HSP10. This would suggest that several molecules similar to ATP can induce this association, such as the mizoribine.  Are other imidazole molecules able to have the same effect? If ATP induces the complex also ATP can inhibit IL6 production? Can ATP, in the experimental system used by the authors, increase the complex?  How much the "novel" immunosuppressive effect is relevant biologically? 

Only IL6 is inhibited? Does mizoribine inhibit the release in culture supernatant of IL6? 

It appears that the concentrations reached in vivo are not ones that work in vitro on cells (discussion section). Is this correct? Please explain better.

Minor points

line 118 Mizorivine should be Mizoribine

Comments on the Quality of English Language

English language is good.

Author Response

Comments and Suggestions for Authors

Thank you very much for reviewing our manuscript and for your valuable comments. We have sincerely responded to your points.

Suggestions for Authors

This work deals with the relevance of mizoribine in promoting the complex formation of the HSP60 with HSP10. Also, this compound can inhibit the production of IL6 in the cell culture system composed of Caco2 and THP1 cells.

The findings are of interest. Mizoribine is an immunosuppressive agent described several years ago. The effect on IL6 is already known for a long time. The authors stated that a novel mechanism of immune suppression mediated by mizoribine has been identified. Indeed, high concentrations of mizoribine promoted HSP60-HSP10 complex formation and consequently suppressed IL-6 expression.

This is the most important finding of this work. In other words, the mizoribine appears to favor a complex among the two different HSPs that reduces the production of IL6.

In the abstract, the authors stated that ATP induces the complex HSP60-HSP10. This would suggest that several molecules similar to ATP can induce this association, such as the mizoribine.  Are other imidazole molecules able to have the same effect? If ATP induces the complex also ATP can inhibit IL6 production? Can ATP, in the experimental system used by the authors, increase the complex?  How much the "novel" immunosuppressive effect is relevant biologically?

As to whether ATP can also inhibit IL6 production, we would like to test this in the future as a new study. The novel immunosuppressive effect of HSP60 has been previously reported to be related to immunity, but the promotion of molecular chaperone complex formation by mizoribine is a discovery and is considered biologically important.

Only IL6 is inhibited? Does mizoribine inhibit the release in culture supernatant of IL6?

This study analyzed the inhibitory effect of IL6 alone? We will analyze whether mizoribine inhibits the release of IL6 in culture supernatants and, in due course, for other cytokines.

It appears that the concentrations reached in vivo are not ones that work in vitro on cells (discussion section). Is this correct? Please explain better.

I described this point in the Discussion section, as suggested.

Minor points

line 118 Mizorivine should be Mizoribine.

We corrected it, as suggested.

Text corrections are in red.

Reviewer 4 Report

Comments and Suggestions for Authors
  1. In this manuscript, Miura et. al have explored the critical role of antimetabolite, Mizoribine (MZR), an immunosuppressive drug developed from the fungus Eupenicillium brefeldianum. This is an interesting manuscript, however there are several areas that need improvement:
  2. The authors need to add statistics and the usage of appropriate statistical tests to compare the results of their findings.
  3. The authors need to improve the presentation of all the results specifically in reference to statistical tests - "Figure 4. Effects of Mizoribine on induction of IL-6. ) (B). PBS or Mizoribine was 111 added to cells for 0, 3, and 6 hours to a final concentration of 10 mM. The THP-1 cell lysate was 112 analyzed by SDS-PAGE following immunoblotting using antibodies against IL-6 and β-actin. IL-6 113 and b-actin were circled by red rectangles  (C). The IL-6/b-actin ratio based on (C) was quantified 114 using ChemiDoc XRS. The signal in each lane of the immunoblot in (C) was quantified and graphed. 115 The data are expressed as mean ± SD (n=3) (D)."
  4. The authors need to include a section on statistical rigor and design of the experiment.
  5. The author can tabulate the Mizoribine related publications and their findings so that a comparison can be made of their clinical roles in the discussion.
  6. The authors need to expand on only including IL-6 in their analysis while other cytokines, chemokines, and cellular properties are equally important in the assessment.
  7. The authors need to expand the introduction section to better introduce the relevance of chaperones and their investigation in the study.
  8. The manuscript would benefit from the strength and limitation section that can be utilized to discuss the drawbacks of the study and future directions.
Comments on the Quality of English Language

English needs to be improved for better flow.

Author Response

Comments and Suggestions for Authors

Thank you very much for reviewing our manuscript and for your valuable comments. We have sincerely responded to your points.

Suggestions for Authors

  1. In this manuscript, Miura et. al have explored the critical role of antimetabolite, Mizoribine (MZR), an immunosuppressive drug developed from the fungus Eupenicillium brefeldianum. This is an interesting manuscript, however there are several areas that need improvement:
  2. The authors need to add statistics and the usage of appropriate statistical tests to compare the results of their findings.
  3. The authors need to improve the presentation of all the results specifically in reference to statistical tests - "Figure 4. Effects of Mizoribine on induction of IL-6. ) (B). PBS or Mizoribine was 111 added to cells for 0, 3, and 6 hours to a final concentration of 10 mM. The THP-1 cell lysate was 112 analyzed by SDS-PAGE following immunoblotting using antibodies against IL-6 and β-actin. IL-6 113 and b-actin were circled by red rectangles (C). The IL-6/b-actin ratio based on (C) was quantified 114 using ChemiDoc XRS. The signal in each lane of the immunoblot in (C) was quantified and graphed. 115 The data are expressed as mean ± SD (n=3) (D)."

Regarding comments 2 and 3, we reworked the figure with statistics and appropriate statistical tests (Figs. 3D, 4B, and 4D) as suggested.

  1. The authors need to include a section on statistical rigor and design of the experiment.

We have added sections on statistical rigor and experimental design in the “Materials and Methods” section as suggested.

  1. The author can tabulate the Mizoribine related publications and their findings so that a comparison can be made of their clinical roles in the discussion.

We thought about the clinical role in the Discussion section as suggested.

  1. The authors need to expand on only including IL-6 in their analysis while other cytokines, chemokines, and cellular properties are equally important in the assessment.

As to why we chose only IL-6 for this study, we discussed it in the Introduction and Discussion section as suggested.

  1. The authors need to expand the introduction section to better introduce the relevance of chaperones and their investigation in the study.

We have expanded the introductory section as suggested.

  1. The manuscript would benefit from the strength and limitation section that can be utilized to discuss the drawbacks of the study and future directions.

The study of a new mechanism of action of Mizoribine concerning the production of molecular chaperones and IL-6 is a strength.

Text corrections are in red.

Round 2

Reviewer 1 Report

Comments and Suggestions for Authors

The manuscript should be accepted for short publication. 

Reviewer 3 Report

Comments and Suggestions for Authors

The authors replied partially to reviewer's queries.  The analysis of inhibition of other cytokines than IL has not been performed (at least, I do not see results on this point). Thus, it is not clear whether the effect is just on IL6 and how much the Mizoribine can indeed inhibit immune response in general and what kind of immune response. 

Comments on the Quality of English Language

English is good.